# Strongly coupled magneto-exciton condensates in large-angle twisted double bilayer graphene

Qingxin Li[1,6], Yiwei Chen[1,6], LingNan Wei[1,6], Hong Chen[1,6], Yan Huang[1], Yujian Zhu[1], Wang Zhu[1], Dongdong An[1], Junwei Song[1], Qikang Gan[1], Qi Zhang[1], Kenji Watanabe [2], Takashi Taniguchi [3], Xiaoyang Shi[4] ✉, Kostya S. Novoselov [5], Rui Wang[1] ✉, Geliang Yu [1] ✉ & Lei Wang [1] ✉

Excitons, pairs of electrons and holes, undergo a Bose-Einstein condensation at low temperatures. An important platform to study excitons is double-layer two-dimensional electron gases, with two parallel planes of electrons and holes separated by a thin insulating layer. Lowering this separation ($d$) strengthens the exciton binding energy, however, leads to the undesired interlayer tunneling, resulting in annihilation of excitons. Here, we report the observation of a sequences of robust exciton condensates (ECs) in double bilayer graphene twisted to ~10° with no insulating mid-layer. The large momentum mismatch between two graphene layers suppresses interlayer tunneling, reaching a $d$ ~ 0.334 nm. Measuring the bulk and edge transport, we find incompressible states corresponding to ECs when both layers are in half-filled $N = 0, 1$ Landau levels (LLs). Theoretical calculations suggest that the low-energy charged excitation of ECs can be meron-antimeron or particle-hole pair, which relies on both LL index and carrier type. Our results establish a novel platform with extreme coupling strength for studying quantum bosonic phase.

An EC is a Bose-Einstein condensate formed when a large number of electron-hole pairs occupy the ground state with macroscopic phase coherence[1]. In bulk materials, condensed excitons can be generated by optical pumping but with short lifetimes[2]. In small-bandgap semiconductors and semimetals, ECs are predicted to live for longer time whereby exciton binding energy exceeds the charge gap[3]. But the structural character of spontaneous symmetry breaking in these solid-state systems may hamper the possibility to realize superfluidity[4,5]. Double-layer system subject to finite magnetic field is shown as impressive platform for exciton condensation[6,7]. As the recombination is blocked by midlayer insulator, electron-like carriers in a partially filled LL in one layer couple with hole-like carriers in the other, forming interlayer magneto-excitons, which then experience a Bose-Einstein condensation to a coherent superfluid ground state[8–11].

The energy of the magneto-excitons is determined by the ratio of intralayer and the interlayer Coulomb interaction: $E_{intra}/E_{inter} \sim d/l_B$, where $l_B = \sqrt{\hbar/eB}$ is the magnetic length, $\hbar$ is the reduced Planck constant, $e$ is the electron charge, and $B$ is magnetic field. An attractive characteristic of such quantum hall bilayer structure is that $d/l_B$ can be tuned by $B$ and $d$, providing an opportunity to adjust the electron-hole

[1]National Laboratory of Solid-State Microstructures, School of Physics, Nanjing University, Nanjing 210093, China. [2]Research Center for Electronic and Optical Materials, National Institute for Materials Science, 1-1 Namiki, Tsukuba 305-0044, Japan. [3]Research Center for Materials Nanoarchitectonics, National Institute for Materials Science, 1-1 Namiki, Tsukuba 305-0044, Japan. [4]Environmental and Sustainable Engineering, College of Engineering and Applied Science, University at Albany, Albany, NY 12222, USA. [5]Institute for Functional Intelligent Materials, National University of Singapore, Building S9, 4 Science Drive 2, Singapore 117544, Singapore. [6]These authors contributed equally: Qingxin Li, Yiwei Chen, LingNan Wei, Hong Chen. ✉ e-mail: xshi7@albany.edu; rwang89@nju.edu.cn; yugeliang@nju.edu.cn; leiwang@nju.edu.cn

coupling strength and the average distance between excitons. In this way, it facilitates exploring quantum condensate phase changes in bosonic system, e.g., the crossover of weak-coupling Bardeen-Cooper-Schrieffer (BCS) pairing to a strong-coupling Bose-Einstein condensation pairing[12] and the transition of superfluid coherent phase to translational symmetry breaking supersolid coherent phase[13]. However, the intriguing region with extremely strong coupling, which needs tiny $d/l_B$ remains elusive due to the difficulty in achieving extremely small $d$ without raising interlayer tunneling.

Recently, progress on reducing the $d$ down to subnanometer has been made in natural bilayer WSe$_2$, where interlayer tunneling was avoided by the intrinsic spin-valley structure[14]. However, the unipolar nature of such semiconductor system limits the observation of magneto-excitons only on the hole side. Another candidate reaching such a small $d$ is the large-angle twisted graphene system, where a large momentum mismatch between different sheets suppresses the interlayer tunneling[15], making it possible to realize ECs in the strong coupling limit with layers widely covering both the electron and hole sides. Although recent studies have shown some plausible traces of ECs in such twisted bilayers by observing quantum Hall states(QHSs) at some incomplete odd-integer total fillings limited on one carrier side[16,17], due to contact quality issues and disorders[16], the sequence of ECs is yet to be observed.

Here, we reported the observation of a complete sequence of ECs emerging at both electron and hole fillings with extremely strong coupling strength in high-quality large twisted angle twisted double bilayer graphene(TDBG) devices (Fig. 1a). At finite magnetic field, the interlayer tunneling gap due to spatial wave functions overlap in two bilayers can be negligible($10^{-10}$meV) based on our numerical calculation(see Supplementary Note 3). By measuring the bulk and edge transport properties, we unambiguously identified these robust ECs which appear at the half-filled $N = 0$ and $N = 1$ LL. Thermally activated

measurements combined with theoretical models indicate that the low-energy charged excitation of ECs is topologically nontrivial spin-texture in $N = 0$ LL[18], whereas for $N = 1$ LL, it changes from such spin-texture on the hole side to particle-hole pair on the electron side.

## Results

We fabricated high-quality TDBG devices with 'cut and pick-up' transfer method[19,20] by picking up and twisting two pieces of bilayer graphene, cut from a single flake, to an angle about 10°. (see in Methods). Depicted in Fig. 1b, the structure of the device, which contains top and bottom graphite gates with voltages $V_T$ and $V_B$, allows us to independently tune carrier density: $n = (C_B V_B + C_T V_T)/e$, and displacement field: $D = (C_B V_B - C_T V_T)/2$, where $C_T(C_B)$ is top(bottom) gate capacitance and $e$ is the elementary charge. Figure 1c shows the longitudinal resistance $R_{xx}$ as a function of $n$ and $D$ in the absence of a magnetic field. A high resistance state appearing around zero values of $n$ and $D$ suggests the presence of crystal fields[15], which occurs due to the imbalance of electron occupancy between the outer two layers and inner ones. Upon increasing the displacement field, the high resistance state evolves into two splitting resistive traces, dividing the diagram for $D > 0$ into five regions (Fig. 1d), which correspond to the different carrier population configurations in the two bilayers. In region I and V, both bilayers are simultaneously populated by either holes or electrons, respectively; in region II and IV, one of the bilayers is gapped while the other is filled; in region III, the two bilayers are populated by opposite carrier types, and the system contains a mixture of holes and electrons. This layer-selective population behavior evidences the two bilayers in our large-angle TDBG are decoupled, allowing the top and bottom gates to control them separately[21,22].(Supplementary Note 1).

Next, we investigate the behavior of the system under magnetic fields. Figure 1f plots the $R_{xx}$ versus the $D$ and total filling factor $\nu_{tot}$ at $B = 14$ T, where $\nu_{tot} = \nu_T + \nu_B$, and $\nu_T$, $\nu_B$ are the LL filling fractions of the

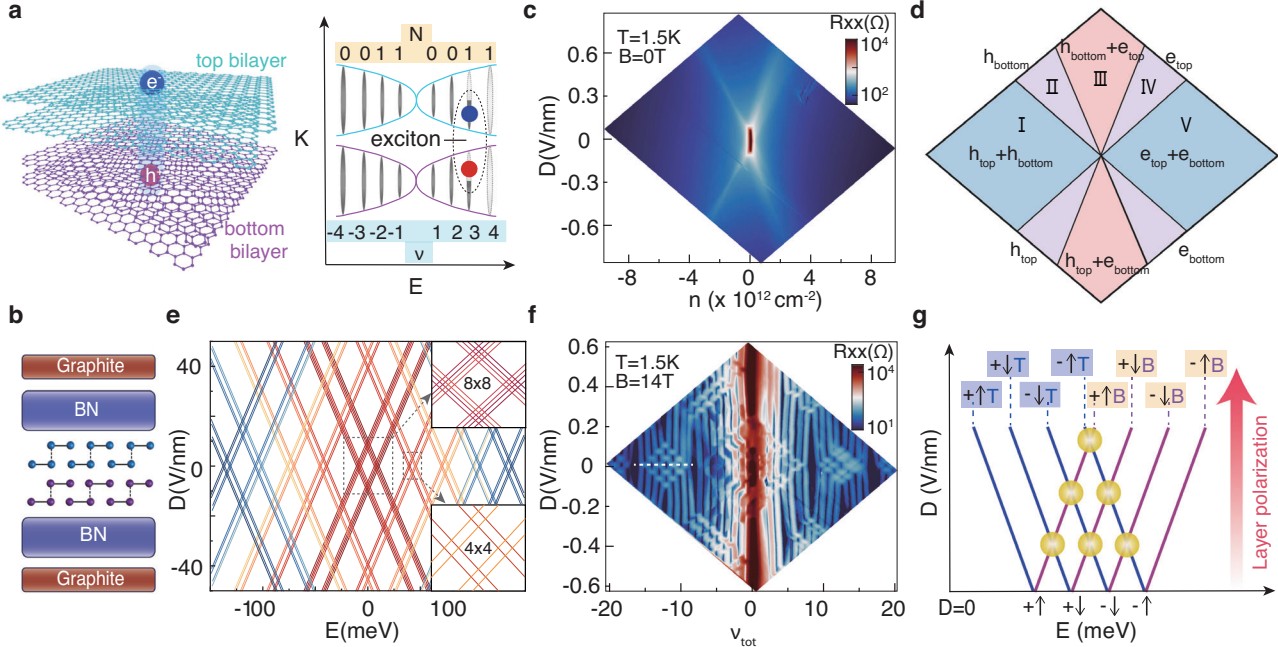

**Fig. 1 | Large-angle TDBG Landau level structure and decoupled carrier population. a** Illustration of magneto-excitons in TDBG. The interlayer tunneling is suppressed by large momentum mismatch, and at finite magnetic field, strong interaction induces electron-like carriers in a half-filled LL in one layer coupling with hole-like carriers in the other, forming interlayer magneto-excitons. **b** Schematic of the TDBG device structure. **c–d** Longitudinal resistance $R_{xx}$ of TDBG at $B = 0$ T and $T = 1.5$ K, versus $n$ and $D$. The map exhibits the different population of charge carriers in two bilayers and the 'layer-targeted' population divides the diagram into

different regions highlighted in **d**. **e** Calculated LL spectra as a function of energy $E$ and $D$ at $B = 14$ T. Inset: Zooming-in on the LL crossings in the 8 × 8 matrix (top) (accidental degeneracy for $N = 0$ and 1) and a typical 4 × 4(bottom) matrix (for $N \geq 2$), respectively. **f** Longitudinal resistance $R_{xx}$ of TDBG at $B = 14$ T and $T = 1.5$ K, versus $\nu_{tot}$ and $D$. The white dashed line marks the four QHSs stabilized by the spin and valley degeneracy lifting at $D = 0$ V/nm in non-zero LL. **g** Schematic illustration of LL crossings driven by displacement field under a constant magnetic field. 'T': top bilayer; 'B': bottom bilayer.

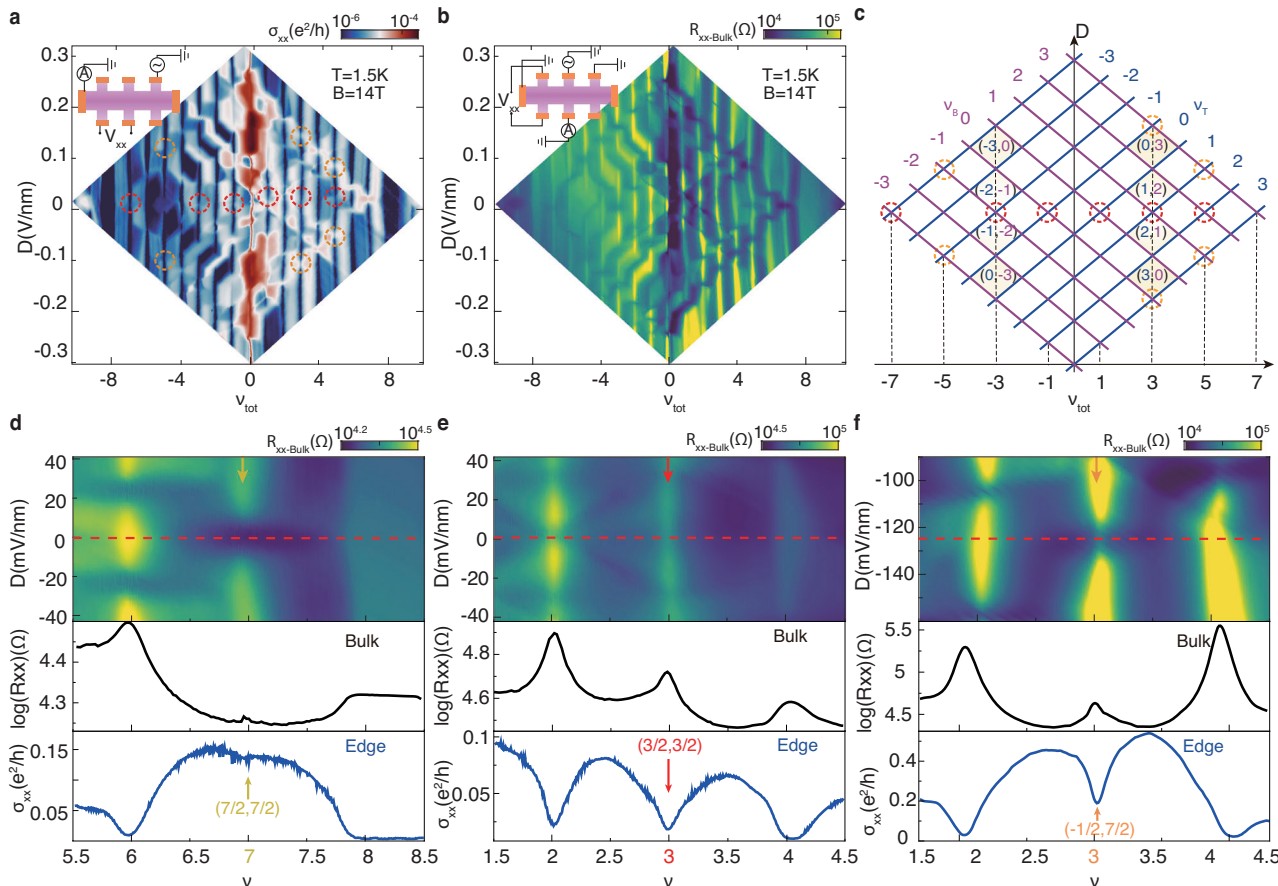

**Fig. 2 | ECs in the zero-energy Landau level matrix. a, b** The longitudinal conductance $\sigma_{xx}$ (**a**) and bulk resistance $R_{xx-Bulk}$ (**b**) versus $D$ and $v_{tot}$ at $B = 14$ T and $T = 1.5$ K for $-9 < v_{tot} < 9$. Dotted red circles(**a**) and orange circles mark the LL crossings manifesting as anomalous conductivity minimum states at $D = 0$ and $D \neq 0$, respectively. The insets show the measurement configurations. To measure $R_{xx-Bulk}$, two contacts are grounded to make sure the signal comes from the bulk instead of edge resistance of the sample. **c,** Schematic LL diagram for $-8 < v_{tot} < 8$. These LL crossings originate from the cross of zero-energy LLs octet of two decoupled bilayers which are marked by different colours. Yellow shades mark two typical IQH regions with the filling factor marked as $(v_T, v_B)$. **d–f** Top panel displays enlarge images of $R_{xx-Bulk}$ versus $v_{tot}$ and $D$ at $B = 14$ T around the three categories LL crossings illustrated in maintext. In each bottom panel, the black curve is the bulk resistance linecut along the red dashed line in the top panel, while the blue curve is the longitudinal conductance linecut in **a**. The yellow, red, and orange arrows point to normal LL crossings, ECs, and less-developed ECs, respectively.

top and bottom bilayer respectively. In bilayer graphene(BLG), the zero-energy LL(ZLL) has eight-fold degeneracy (spin, valley, and accidental orbital degeneracy $N = 1$, $N = 0$), and higher LLs have four-fold degeneracy[23]. As for our system consisting of two decoupled bilayer graphene layers, LLs have an extra two-fold degeneracy corresponding to 'top layer' and 'bottom layer' regulated by $D$. At $B = 14$ T, these degeneracies are fully lifted, showing a sequence of QHSs at all integer fillings as resistance minima lines paralleling to the $n = 0$ line. Strikingly, we observed some repeated '4 × 4' matrices (for $|v_{tot}| > 8$) and a unique '8 × 8' matrix (for $-8 < v_{tot} < 8$). These matrices can be qualitatively understood using the single-particle picture of LL crossings as illustrated in Fig. 1e and g. In a '4 × 4' matrix, along the white dashed line (Fig. 1f) at $D = 0$, four QHSs at even total fillings stabilized by the spin and valley degeneracy lifting. As $|D|$ increases, the layer degeneracy is lifted and each of these four LLs splits into two, subsequently intersect with their neighbors, forming a series of crossing points (marked by yellow circles in Fig. 1g). At these crossing points, the double bilayer system is gapless since both bilayers are in partially filled LLs (more details in Supplementary Note 2).

Now we move on to look into the '8 × 8' matrix centered at the charge neutrality point. In Fig. 2a we plot longitudinal conductance $\sigma_{xx}$ versus $D$ and $v_{tot}$ for $-9 < v_{tot} < 9$. A clear '8 × 8' structural pattern with a forming mechanism analogous to the '4 × 4' matrices is displayed,

which corresponds to crossings of the quantum Hall octet from two decoupled bilayers(schematically illustrated in Fig. 2c). Unexpectedly, focusing on $D = 0$V/nm, we notice a series of exceptional crossing points at $v_{tot} = -7, -3, -1, 1, 3$ and 5(red dotted circles in Fig. 2a and c) manifesting as anomalous states with quantized Hall conductivity and vanishing longitudinal conductivity, which is drastic contrast to the finite conductivity in those normal LL crossings.(see Supplementary Fig. 4). Besides, for $D \neq 0$, a few similar anomalous states also develop with the reduced longitudinal conductivity marked by orange dotted circles in Fig. 2a. These anomalous states are inadequate to be understood from the single-particle LL crossings picture, where the system should show a finite conductivity as both bilayer LLs are half-filled.

In order to investigate the origin of the vanishing of $\sigma_{xx}$ at these crossing points, we further measure the bulk transport properties of these anomalous states using the configuration shown in Fig. 2b inset[24]. Figure 2b maps the bulk resistance $R_{xx-Bulk}$ as a function of $v_{tot}$ and $D$ on the same '8 × 8' matrix. Along $D = 0$ V/nm, each anomalous state manifests a high $R_{xx-Bulk}$ peak, while other LL crossings show $R_{xx-Bulk}$ dips. Based on these two different patterns of $R_{xx-Bulk}$ at the crossing points, we group them into three categories as shown in Fig. 2d, e, f. In the top panels, the center of these $R_{xx-Bulk}(v_{tot}, D)$ maps corresponds to both bilayers are half-filled and we take linecuts at the crossing points

showing $R_{xx-Bulk}$ vs. $\nu_{tot}$ in the bottom panels. For normal crossing points (Fig. 2d), the resistance dip in $R_{xx-Bulk}$ linecut manifests that the system is compressible at these states. This confirms the single-particle LL crossings picture, meanwhile, also indicates the tunneling is negligible between two bilayers otherwise a LL anti-crossing gap would be induced by tunneling[25,26]. On the contrary, for the anomalous crossing points along $D = 0$ (Fig. 2e), the prominent resistive peak demonstrates the state is incompressible, which is beyond the picture of single-particle LL crossing. Given that the tunneling is negligible here, this phenomenon implies the emergence of a correlation energy gap due to many-body interactions. When both bilayers are half-filled, interlayer interactions prompt electrons in one bilayer and holes in the other to form magneto-excitons[10,11] and condense into an incompressible superfluid: exciton condensate. Besides, it's worth pointing out that the crossing points for $D \neq 0$ (Fig. 2f) show comparatively weaker bulk-resistance peak compared to those along $D = 0$. This is presumably due to the imbalanced occupancy of LL orbitals of top and bottom bilayers[13,27] or slight spatial wave functions overlap in two bilayers induced by finite $D$.

To characterize the ECs, we examined the excitation energy gap at all odd-integer filling for $-8 < \nu_{tot} < 8$ with thermal activation measurements. Figure 3a shows the temperature dependence of bulk resistance ($R_{xx-Bulk}$) as a function of $\nu_{tot}$ at $D = 0$ V/nm for $B = 14$ T. The EC gap ($\Delta$) shows an unexpected hierarchy and manifests as a non-monotonic behavior with $\nu_{tot}$ (Fig. 3c). Remarkably, all ECs appear in $-4 < \nu_{tot} \leq 4$ hold obviously larger gap value than those appear in other odd-integer fillings. In BLG, orbital character of the ZLL has been mapped out as the function of filling factors and electric fields[27], and the holes or electrons are fully polarized in a single orbital component ($N = 0$ or 1) covering the whole of accessible parameter space. Based on this picture, we displayed the distribution of orbital index of these two decoupled bilayers with filling factors under strong magnetic field in Fig. 3b. Near $D \approx 0$, LLs of both bilayers with unambiguous orbital index cross with each other, giving rise to the filling sequence of orbital index throughout $-8 \leq \nu_{tot} \leq 8$ as: $-4 < \nu_{tot} \leq 0$, $4 \leq \nu_{tot} \leq 8$ corresponds to $N = 1$ LL and $-8 \leq \nu_{tot} < -4$, $0 \leq \nu_{tot} < 4$ are in accord with $N = 0$ LL. As a result, we find that the EC robustness is tightly associated with LL index and carrier type(electron-hole asymmetry), specifically, ECs appearing within the $N = 1$ orbital of hole side are more stable than those within the $N = 0$ orbital, whereas it is opposite on the electron side.

The orbital wavefunction plays an important role for formation of correlated states. For example, in BLG, compared to the conventional $N = 0$ orbital with sharper composite-fermion interactions, the $N = 1$ orbital has softer interactions which are beneficial for pairing due to an additional node in the single-particle wavefunction[28], and that lead to the observation of the even-denominator fractional QHSs exclusively within LL $N = 1$, not $N = 0$.[29–32]. For our system consisting of two decoupled bilayers, different orbitals may host distinct low-energy excitations, which affects the robustness of ECs. This influence can be better understood by taking pseudospin magnetism picture into consideration in which pseudospin up (down) corresponds to an electron or a hole in the upper(lower) bilayer, and spontaneous-interlayer-coherence broken symmetry occurs as easy-plane pseudo-ferromagnetism[18]. In this case, considering the finite interlayer spacing $d$, topologically stable charged vortices known as meron can emerge.[18,33] Then the merons and anti-merons pairing, leads to the topologically nontrivial spin configurations known as skyrmions. Theoretical works have suggested that the energy of this excitation increases with orbital index $n$[34]. The meron-antimeron spin-texture is not the sole low-energy excitation in the double-layer system. With increasing of LL orbital index, conventional particle-hole pairs may host lower energy due to shorter-ranged interactions caused by excessive screening, and overtakes meron-antimeron pairs. A recent study suggests that in p-type bilayer $WSe_2$ the spin-texture

charged excitation only occurs in LLs $n \leq 2$[14], while higher Landau levels host the particle-hole excitation whose energy decreases with orbital index $n$.

In our TDBG system, theoretical calculations suggest that the low-energy charged excitation of ECs is not only related to LL index but also associated with carrier doping type. Figure 3d shows the theoretical calculation of the $\Delta - N$(LL index) dependence of these two different charged excitations. It reveals that the spin-texture charged excitations for ECs are favoured in the lower LLs on both electron and hole sides, and switch to particle-hole type at higher LL index. However, the transition points between two types of charged excitations are different on the electron and hole sides, with the EC-gap maximum occurring between $N = 1$ and $N = 2$ for hole side and near $N = 0$ for electron side. This discrepancy can be well understood by considering the different screening strength between the hole and electron sides. Being in ZLL of TDBG at $D = 0$, with the total filling increasing, carriers populate QHSs with different spin-valley flavour from $\nu_{tot} = -8$ to 8 sequentially. At the same time, increasing particle density strengthens the screening, driving a shorter-ranged interaction. This, in turn, reduces the excitation energy of particle-hole pair originating from exchange interactions, whereas leaving the spin-texture excitation energy unaffected. This theoretical result well agrees with our observed EC-gap trend. On the hole side, ECs in $N = 1$ and $N = 0$ hold spin-texture excitations, thereby excitation energies monotonically increase with the LL index. Meanwhile, on the electron side, ECs in $N = 1$ with reduced particle-hole pair excitation energy, have smaller gaps compared to the ECs in $N = 0$ with spin-texture excitations.

The evolution of the bulk resistance with $D$ provides further insight into the identification of the two types of charged excitation. Figure 3e plots the bulk resistance as a function of $D$ for ECs at $\nu_{tot}$=1 and 5, corresponding to spin-texture and particle-hole excitation type, respectively. The red curve regions around $D = 0$ mark EC regime, and the system transition into integer quantum hall(IQH) phase(grey curve) with the increasing of $D$. Previous numerical study find that the excitation energy of meron-antimeron pairs show a sharp increase in gap with layer imbalance, while particle-hole excitation energy is independent of the layer imbalance until the zeeman energy exceeds the EC gap[35]. In our system, layer imbalance is regulated by $D$. In this scenario, we find a sharp $R_{xx-Bulk}$ decrease with $D$ in red curve region for $\nu_{tot} = 1$ whereas at $\nu_{tot} = 5$, $R_{xx-Bulk}$ hold a mild response with $D$ (Fig. 3e). This suggests that ECs on electron side in $N = 0$ LL host the spin-texture charged excitation while ECs in $N = 1$ LL have a particle-hole excitation(At $\nu_{tot}$= 1 in $N = 0$ LL, red curve corresponding to meron-antimeron-type ECs with lager slope than gray curve corresponding to IQH, and At $\nu_{tot}$=1 in $N = 1$ LL, red curve corresponding to particle-hole-type ECs with smaller slope than gray curve corresponding to IQH.). On the other hand, on the hole side, ECs in both $N = 0$ and 1 exhibit sharp $R_{xx-Bulk}$ changes, corresponding a spin-texture charged excitation(Supplementary Fig. 8). This result is consistent with our theoretical calculation that the low-energy charged excitation of ECs is different on electron and hole sides.

Finally, to fully identify the nature of ECs, we demonstrate the evolution of ECs with magnetic field. Figure 4a shows $B$ dependence of $R_{xx-Bulk}$ for all the ECs at $D = 0$ V/nm. The $R_{xx-Bulk}$ of all ECs decreases with diminishing of $B$ indicating that our system is in strong coupling regime. In this regime, the main effect of increasing the magnetic field is to raise the excitons density($\propto B$), rather than increasing the $d/l_B (\propto \sqrt{B})$ to soften the exciton pairing strength, which is preferred in weak coupling system[12]. We further find that all ECs gaps are positively correlated with magnetic field and well fit by $\Delta = E_c/E_c(14$ T$)$ (red dashed line in Fig. 4b) which is in line with our numerical calculations (Supplementary Note 4). Both of the particle-hole and the spin-texture excitation energies are related to stiffness ($\rho_s - 1/l_B$), causing the excitation energy is proportional to the Coulomb energy ($E_c = e^2/l_B$)

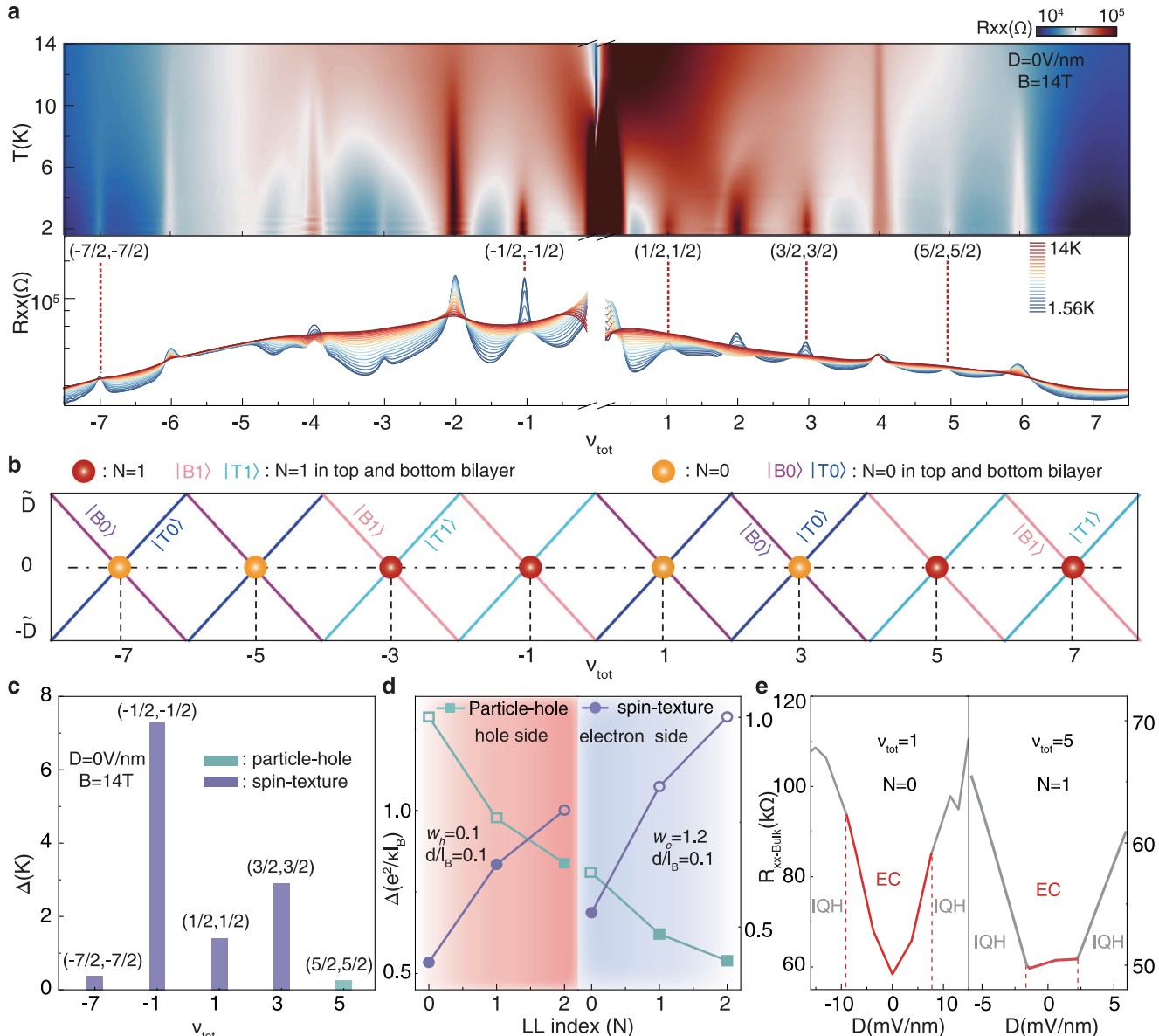

**Fig. 3 | Energy gap of ECs and the low-energy charged excitations.**
**a** Temperature dependence of $R_{xx-Bulk}$ as a function of $v_{tot}$ at $D = 0$ V/nm and $B = 14$ T. Red dotted lines mark the ECs corresponding to both bilayers are half-filled.
**b** Evolution of orbital occupancy($N = 0$, $N = 1$) with total filling factors in decoupled TDBG. **c** Excitation energy gaps of ECs at $B = 14$T and $D = 0$ V/nm. Purple rectangles correspond to ECs with the spin-texture charged excitation and the green rectangle corresponds to EC with the particle-hole charged excitation. **d** Theoretical calculations of energy gap for two types of excitations on hole and electron sides, at $d/l_B$ = 0.1 and $D = 0$ V/nm. Owing to the distinct electron and hole environments, we use the parameter 'w' to quantify the strength of screening effects. Since the screening effect of carriers is weak when filling the zero-energy Landau level starting from hole side, we take $w_h = 0.1$. Conversely, the screening effect is already significant when carriers fill the electron side, so we take the average screening effect for the

electron side as $w_e = 1.2$. The spin-texture excitation energy increases with LL index while particle-hole excitation energy decreases with LL index. Their crossing points for hole and electron sides appear at different positions of LL index due to the screening strength increasing with total filling. The filled markers represent the low-energy excitations of ECs. **e** $R_{xx-Bulk}$ as a function of $D$ for the ECs at $v_{tot}$ = 1, 5, which correspond to $N = 0$ and 1 LL, respectively. Red curve regions mark ECs and grey curves correspond to IQH regime. At $v_{tot}$ = 1 in $N = 0$ LL, red curve corresponding to meron-antimeron-type ECs with lager slope than gray curve corresponding to IQH, and At $v_{tot}$ = 1 in $N = 1$ LL, red curve corresponding to particle-hole-type ECs with smaller slope than gray curve corresponding to IQH. (The rate of change of the bulk resistance with the displacement field near $D = 0$ (the magnitude of the slope) determines the phase boundaries of different phases).

(Supplementary Note 4). Furthermore, it is worth noting that the $N = 1$ orbital in BLG differs from conventional $n = 1$ orbital. In BLG, $N = 1$ orbital contains a combination of both conventional LL orbital $n = 0$ and $n = 1$ wavefunctions distributed on different atomic sites of BLG, with the relative weight of $n = 0$ wavefunction increasing with $B$[31,32]. Hence, under higher magnetic fields beyond our experiments (about $B > 25$ T)[31,32], the extensive participation of $n = 0$ wavefunction in BLG $N = 1$ orbital renders ECs in decoupled bilayers prone to hold excitation energy deviating from the trend of $E_c$.

## Discussion

In summary, we have experimentally observed remarkable magneto-excitons and their EC phase in ZLL region of large-angle twisted TDBG. Interlayer tunneling is suppressed by large momentum mismatch, and we demonstrate the ECs in the strong coupling limit with sub-nanometer atomic separation between the two bilayers. The different carrier screening strengths in electron and hole sides lead to distinct stability of ECs in both carrier types, and the evolution of ECs with LL index unveiled a change of the low-energy charged excitation from

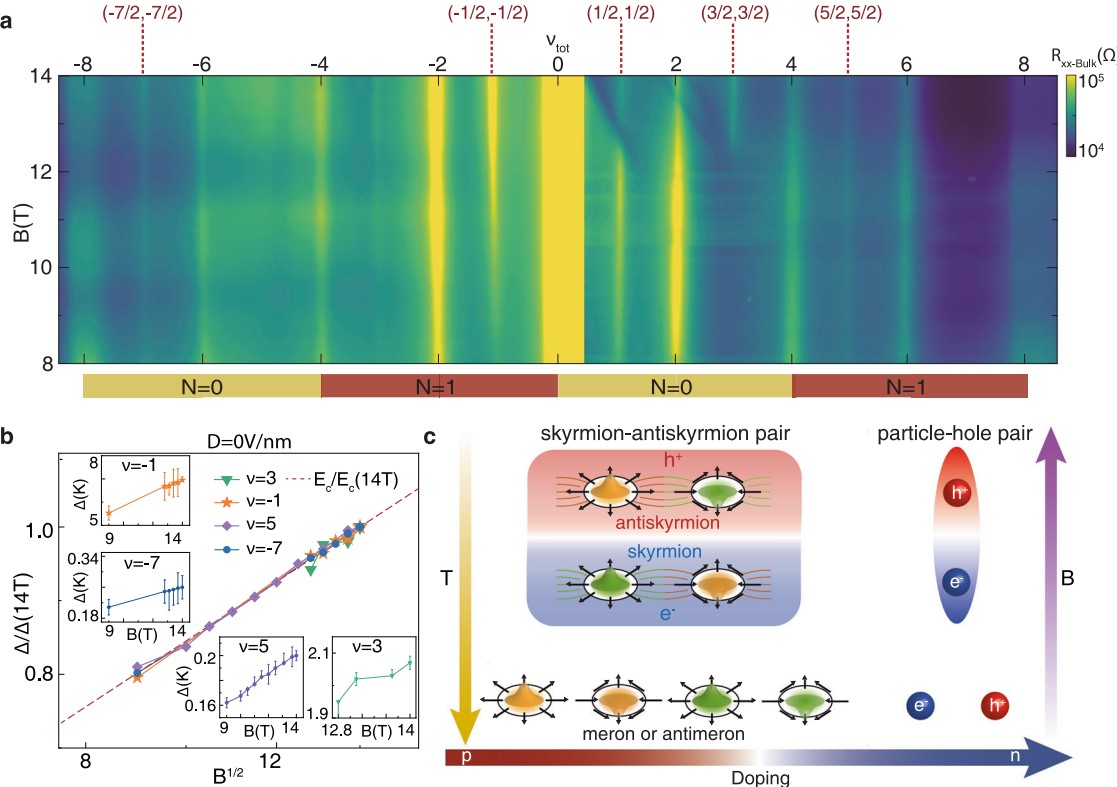

**Fig. 4 | Evolution of ECs with magnetic field. a** The $R_{xx-Bulk}$ versus $B$ and $v_{tot}$ at $D =$ 0 V/nm and $T = 1.55$ K for $-9 < v_{tot} < 9$. **b** Scaled activation gap $\Delta/\Delta(14$ T$)$ of ECs at different total fillings as a function of $\sqrt{B}$ for 9 T $\leq B \leq$ 14 T. The red dashed curve corresponds to the scaled Coulomb energy, $E_c/E_c(14$T$)$ ($\propto \sqrt{B}$). Insets: details of activation gaps of each EC at different magnetic fields, error bars in the gap measurement arise primarily from ambiguity in choosing the thermally activated region. **c** Illustration of the low-energy charged excitations of ECs and their evolution with tuning temperature($T$), magnetic field($B$) and carrier density. The

bottom panel shows four types of merons(antimerons) (left) and free electrons and holes(right). A meron and an antimeron have opposite vorticity, but carry the same electric charge($\pm e/2$) and they can pair to form skyrmion or antiskyrmion which both carry unit of topological charge($\pm e$) but with opposite charge. (shown in top left panel). The spin-texture(particle-hole) EC ground states form by pairing skyrmions and anti-skyrmions (electrons and holes) at low temperature and finite magnetic field. Increasing doping would make the low-energy charged excitation of ECs change from spin-texture type to particle-hole type.

meron-antimeron pair to particle-hole pair on different carrier doping types. The variations in pairing behavior concerning magnetic field, doping, and temperature are summarized in Fig. 4c. Using electrostatic gating, thus we achieved unprecedentedly modulating the topology of low-energy charged excitation of ECs, providing further opportunities on application of skyrmion-type devices in magnetic data storage and topological quantum computing. Moreover, the signature of incompressible states in finite displacement field may lead to an unconventional route to explore non-equilibrium ECs[36] or multi-polar excitonic[37] physics.

## Methods
### Sample fabrication and measurement setup
The three devices (device A - C) in our text were all fabricated using the 'cut-and-stack' technique[19]. The raw materials for the preparation of each device, hexagonal boron nitride(hBN)(about 30nm), graphite and bilayer graphene are obtained from mechanically exfoliation onto Si/SiO$_2$ substrate. Their thickness and quality were then identified by optical microscopy and atomic force microscopy. Before stacking, we first cut the bilayer graphene into two pieces using atomic force microscopy. Then we used hBN, grphite and precut bilayer graphene pieces to assembled the graphite/BN/TDBG/BN/grphite stack using dry pick-up technique with a stamp consisting of polypropylene carbonate(PPC) film and polydimethylsiloxane(PDMS). Using graphite as the gate above and below the TDBG reduces the disorder and defects introduced during evaporation compared to metal gates. The stack is then annealed under high vacuum at 400 °C for 25 minutes. Next we

defined the geometry of the topgate and hall bar by CHF$_3$/O$_2$ etching. Finally, electrode contact was evaporated with Cr/Pd/Au (1/15/100nm) metal by e-beam evaporation.

Transport measurements were carried out in cryogenic superconducting magnets with base temperature of 1.5 K. The four-terminal resistance were measured using low-frequency lock-in techniques at 17.777Hz with a current excitation of 20 nA.

## Data availability
The data that support the findings of this study are available from the corresponding authors upon request.

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

## Acknowledgements

L.W. acknowledges the National Key Projects for Research and Development of China (Grant Nos. 2022YFA120470, 2021YFA1400400), National Natural Science Foundation of China (Grant No. 12074173), Natural Science Foundation of Jiangsu Province (Grant No. BK20220066) and Program for Innovative Talents and Entrepreneur in Jiangsu(Grant No.JSSCTD202101). K.W. and T.T. acknowledge support from the JSPS KAKENHI (Grant Numbers 20H00354 and 23H02052) and A3 Foresight by JSPS. K.S.N. is grateful to the Ministry of Education, Singapore (Research Centre of Excellence award to the Institute for Functional Intelligent Materials, I-FIM, project No. EDUNC-33-18-279-V12) and to the Royal Society (UK, grant number RSRP R 190000) for support. G.Y. acknowledges the National Natural Science Foundation of China (Grant No. 11974169) and the Natural Science Foundation of Jiangsu Province (Grant No. BK20220066).

## Author contributions

L.W. and Q.L. conceived the experiment. Q.L., Y.C., L.N.W., H.Y., Z.Y., Z.W., A.D., J.S., Q.G. and Q.Z. fabricated the samples. Q.L., Y.C., L.W., K.S.N. and G.Y. performed transport measurement and data analysis. R.W., H.C. and S.X. performed theoretical calculations. K.W. and T.T. supplied hBN crystals. Q.L., Y.C., H.C. and L.W. wrote the manuscript with input from all co-authors.

## Competing interests

The authors declare no competing interest.
