## [Peer Review File · Nature Communications]

REVIEWER COMMENTS

Reviewer #1 (Remarks to the Author):

This manuscript reports on evidence of exciton condensates (EC) formed in the LLs of a large-angle double-twisted bilayer graphene system through transport measurements. It highlights how the stability of the EC depends on the LL orbital number and carrier type. I have the following questions/comments before I can recommend the manuscript for publication:

1. The authors emphasize the stability of EC on carrier type. This conclusion may be misleading, because the effect of the carrier type in this specific system cannot be generalized to other systems. Instead, according to the discussion in the manuscript, the differences in the electron and hole side are rather because of different screening strengths due to different carrier densities. The effect of the screening strength (rather than the carrier type) is a parameter that can be generalized to other systems.

Related, the critical parameter in the numerical studies that reproduces the differences on the electron and hole side (Fig.3d), is w/l_B . However, I couldn't find what is w here. The definition of this parameter and how the values (0.1 vs 1.2) are chosen for the hole and electron side is critical and should be discussed.

2. The authors emphasize the stability of EC on the orbital number and carrier type, but I didn't find any discussion to address the difference for, e.g. $\nu = -3$ and -1 , which are both composed of $N = 1$ orbitals and on the hole side. This suggests that there are more relevant parameters in addition to what is addressed in the manuscript.

3. The idea of making EC in large-twist-angle graphene systems is not new, and similar results have been obtained in twisted bilayer graphene, e.g. Ref. 16, 17. Here the authors replace the monolayer graphene with bilayer graphene, and I'm not convinced that this difference brings in enough advancement for publication in nature communications.

Reviewer #2 (Remarks to the Author):

The manuscript is well written and the data is of high quality. The authors conduct a comprehensive study on magneto-exciton condensates in large-angle twisted double layer graphene. The longitudinal resistance at 14T as the function of total filling factor and displacement field undoubtedly show the Landau level structure in the decoupled regime, which means a very small tunneling strength that helps the formation of excitons. Interestingly, the longitudinal conductance and the bulk resistance show a sequence of dips and peaks at the odd filling factors at zero displacement field, respectively. That means there are some gapped states in these filling factors, indicating the possible formation of magneto-exciton condensates. The observation of the sequence of magneto-exciton condensates is quite novel, since the magneto-exciton condensates can only be found at total filling factor in GaAs system. Moreover, the author analyzes the gaps of the sequence quantitatively, which are well consistent with theory. Although this is certainly an interesting work, some issues are still needed to be addressed before publication in Nature Communication. 1. The authors mention that 10° twisted angle helps to reduce the tunneling strength because of momentum mismatch. Other large twisted angles can also make momentum mismatch in two layers. Are there some similar sequences of states observed in twisted double bilayer graphene with other large twisted angles? 2. What's the reason that magneto-exciton condensates are not observed in some odd filling factors? For example, there are no magneto-exciton condensates at and in the hole regime. Especially, it is contradictory to the theory that the system should have a large gap at since the orbital occupancy is $\nu = 0.5$. In Figure 3e, the authors show R_{xx} -Bulk as a function of D to illustrate the evolution from exciton condensates to integer quantum Hall states where the red line shows the phase boundary. However, the clarification of how to define the boundary should be added in the main text, which may help the readers. 4. The axis label of Figure 3e should be changed from R_{xx} -bulk.

Reviewer #3 (Remarks to the Author):

This work reports the result of experimental transport studies and theoretical calculations of the twisted bilayer graphene heterostructure with the graphene layers aligned at a high angle. Similar heterostructures were explored before, however, they either have a spacer between the graphene layers, which leads to large separation between the layers and lower interlayer interactions or are unipolar in nature and only report observations on either hole or electron side.

In my view, the topic manuscript is suitable for Nature Communications as it reports on the observation of an exciton condensate in a novel two-dimensional electronic system with ultrastrong interlayer interactions due to the absence of a spacer layer. While authors cite several other works that study the same type of heterostructure, the authors are the first to observe a complete sequence of a strong Bose-Einstein exciton condensate at both electron and hole fillings. A few remarks:

1. What allows the authors to study the complete sequence of EC on both electron and hole fillings as opposed to the research done in references [16] and [17]? Is it a better quality of devices or a difference in the measurement technique? A discussion explaining the cause of the differences in [16,17] and the authors work would be very instructive.

2. In the description of Figure 1f, the x axis is described as n while the figure itself shows v_{tot}

Responses to the reviewers:

We sincerely thank the reviewers for taking the time to assess our manuscript and raising suggestions to improve it. We believe that this letter and the revised manuscript fully addressed their questions and comments. In the following, we respond in detail to the questions and comments that were raised one by one. Throughout this letter we used blue color for the questions by the reviewers and black color for our responses. Changes made in the revised manuscript are highlighted in yellow.

Reviewer #1 (Remarks to the Author):

This manuscript reports on evidence of exciton condensates (EC) formed in the LLs of a large-angle double-twisted bilayer graphene system through transport measurements. It highlights how the stability of the EC depends on the LL orbital number and carrier type. I have the following questions/comments before I can recommend the manuscript for publication:

We thank the reviewer for carefully reading and summarizing the manuscript. We also appreciate the detailed comments/questions that help to improve our manuscript. We address the comments in detail below.

1. The authors emphasize the stability of EC on carrier type. This conclusion may be misleading, because the effect of the carrier type in this specific system cannot be generalized to other systems. Instead, according to the discussion in the manuscript, the differences in the electron and hole side are rather because of different screening strengths due to different carrier densities. The effect of the screening strength (rather than the carrier type) is a parameter that can be generalized to other systems.

We agree with the reviewer that the difference between the results obtained from electron and hole sides is attributed to differences in the screening strength. In the conclusion section of the main text, we have modified relevant descriptions to emphasize this point as following: 'The different carrier screening strengths in electron and hole sides lead to distinct stability of ECs in both carrier types and the evolution of ECs with LL index unveiled a change of the low-energy charged excitation from meron-antimeron pair to particle-hole pair on different carrier doping types.'

Related, the critical parameter in the numerical studies that reproduces the differences on the electron and hole side (Fig.3d), is w/l_B . However, I couldn't find what is w here. The definition of this parameter and how the values (0.1 vs 1.2) are chosen for the hole and electron side is critical and should be discussed.

We thank the reviewer for the suggestion. The definition of screening strength parameter ' w ' is shown in the supplementary information in page 6. To make it clearer for readers, we have added a description about screening strength parameter and why we choose 0.1 and 1.2 for the hole side and the electron side in Fig. 3d caption, in addition, we have added a detailed explanation in the supplementary information part.

‘Owing to the distinct electron and hole environments, we use the parameter ‘ w ’ to quantify the strength of screening effects. Since the screening effect of carriers is weak when filling the zero-energy Landau level starting from hole side, we take $w_h = 0.1$. Conversely, the screening effect is already significant when carriers fill the electron side, so we take the average screening effect for electron side as $w_e = 1.2$.’

In our system, being in zero-energy Landau level of TDBG at $D = 0$, with the total filling increasing, carriers populate quantum Hall states with different spin-valley flavour from $\nu_{\text{tot}} = -8$ to 8 sequentially. At the same time, increasing particle density strengthens the screening, driving a shorter-ranged interaction. This, in turn, reduces the excitation energy of particle-hole pair originating from exchange interactions, whereas leaving the spin-texture excitation energy unaffected.

We use the parameter ‘ w ’ to quantify the strength of screening effects. In the particle-hole excitation, the gap of ECs can be expressed as follows:

$$\Delta E \approx \frac{1}{4\pi} \int d^2\mathbf{k} (V_{tt}(k) + V_{bb}(k)) [L_N(\frac{l_B^2 k^2}{2})]^2 e^{-\frac{l_B^2 k^2}{2}}$$

We model the electron side Coulomb interaction as $2\pi e^{-w_e q l_B} / |q|$, while the hole side potential is $2\pi e^{-w_h q l_B} / |q|$, here the parameter ‘ w_e ’ and ‘ w_h ’ quantify the strength of screening effects of electron and hole side (PHYSICAL REVIEW B 107, 165427 (2023), PHYSICAL REVIEW C 71, 054005 (2005)). Since the screening effect of carriers is weak when filling the zero-energy Landau level starting from hole side, we take $w_h = 0.1$. Conversely, the screening effect is already significant when carriers fill the electron side, so we take the average screening effect for electron side as $w_e = 1.2$.

2. The authors emphasize the stability of EC on the orbital number and carrier type, but I didn't find any discussion to address the difference for, e.g. $\nu = -3$ and -1 , which are both composed of $N = 1$ orbitals and on the hole side. This suggests that there are more relevant parameters in addition to what is addressed in the manuscript.

There seems a difference between $\nu = -3$ and -1 . In device A, the state at $\nu = -1$ has both longitudinal conductance dip, bulk resistance peak and a well-developed gap, corresponding to exciton condensate (Fig 3c). At $\nu = -3$, we observed a longitudinal conductance dip and a bulk resistance peak (see Fig. 2a and Fig. 2b). Our theory predicts the presence of gapped EC here, yet experimentally, we did not observe a well-developed gap at 1.55K in device A. We attribute this to some possible local unevenness. The temperature dependence curves actually show an onset of a trend towards EC gap formation at a base temperature of 1.55K (Fig. R1a below). At lower temperatures (which can suppress disorder effects) or under stronger magnetic field (which can provide a larger interaction energy scale and enhances the EC gap that overcomes disorder effects), we may observe EC at $\nu = -3$.

Due to device A is destroyed during measurement, we made another device included in FIG. S3d, the hole side of this new device revealed well-developed ECs at $\nu = -3$ with a distinct gap. (also see Fig. R1b, c below).

Fig. R1. (a). Temperature dependence of R_{xx} -Bulk as a function of ν_{tot} at $D = 0$ V/nm and $B = 14$ T in device A. At $\nu_{tot} = -3$, the temperature dependence curves actually show an onset of a trend towards EC gap formation at a base temperature of 1.55K. **(b).** Temperature dependence of longitudinal conductance σ_{xx} as a function of ν_{tot} at $D = 0$ V/nm and $B = 14$ T in device C. The state at $\nu_{tot} = -3$ has a well-developed EC gap shown in **(c)**.

3. The idea of making EC in large-twist-angle graphene systems is not new, and similar results have been obtained in twisted bilayer graphene, e.g. Ref. 16, 17. Here the authors replace the monolayer graphene with bilayer graphene, and I'm not convinced that this difference brings in enough advancement for publication in nature communications.

Although previous studies examined the large-angle twisted graphene systems, the complete excitonic sequence has not been observed due to device quality limitations (as stated in the maintext in Ref.16 : ‘...the interaction-related ground states appear varies among devices mainly related to contact quality issues and specific disorder details.’).

Our high-quality devices enabled us, for the first time, to observe a complete sequences of robust exciton condensates (ECs) with extremely strong coupling strength in large-angle twisted double bilayer graphene and investigate the corresponding low-energy charged excitation modes of ECs. Our work presents novelty and holds unique significance, which are also pointed out by reviewers #2 and #3. We list the novelty and significance of this work as below.

- 1) We observed a complete sequence of magneto-exciton condensates emerging at both electron and hole doping sides in the $N = 0$ and $N = 1$ Landau levels (LLs), for the first time.

Ref.17 (twisted bilayer graphene) only reports some interlayer-coherent quantum Hall states and suggests that large angle twisted graphene system is a platform to realize exciton condensates. However, ref. 17 does NOT make a claim on any exciton condensates in their work. Besides, the interlayer-coherent states only appear on $N = 1$ Landau level, $N=0$ is missing.

Ref.16 claims some plausible traces of exciton condensates only appearing on hole carrier side and only for $N = 1$ Landau level at $\nu_{tot} = -1, -3$.

Our work reports a complete sequence of magneto-exciton condensates emerging at both electron and hole doping sides in the $N = 0$ and $N = 1$ Landau levels (LLs) at $\nu_{tot} = -7, -$

3, -1, 1, 3, 5. The complete sequence of magneto-exciton condensates in our system allows us to characterize the exciton condensates fully and further investigate the lowest-energy charged excitations of those exciton condensates.

- 2) In contrast to ref. 16, we compared the observed magneto-exciton condensates with our theoretical model and obtained conclusions that the low-energy charged excitation of ECs can be meron-antimeron or particle-hole pair, which relies on both LL index and carrier type.

Besides, the evolution of the bulk resistance with D in our work provides further insight into the identification of the two types of charged excitation. (Fig. 3e in main text).

These results show large-angle twisted bilayers as an experimental platform with extreme coupling strength for studying quantum bosonic phase and its low-energy excitations.

- 3) In contrast to interlayer-coherent states in twisted bilayer graphene in ref. 17, our twisted bilayer-bilayer graphene system is very different and we show TDBG system provides a better platform for studying correlated excitonic states.

The bilayer graphene has the potentially more favourable electronic dispersion in comparison with monolayer, and the zeroth Landau level of bilayer is eightfold degenerate, with the spin and valley isospin degeneracy supplemented by an accidental orbital degeneracy. This multitude of broken symmetry states further expands the phase diagram of possible superfluid states and gives opportunity to study a complete sequence of magneto-exciton condensates.

Reviewer #2 (Remarks to the Author):

The manuscript is well written and the data is of high quality. The authors conduct a comprehensive study on magneto-exciton condensates in large-angle twisted double layer graphene. The longitudinal resistance at 14T as the function of total filling factor and displacement field undoubtedly show the Landau level structure in the decoupled regime, which means a very small tunneling strength that helps the formation of excitons. Interestingly, the longitudinal conductance and the bulk resistance show a sequence of dips and peaks at the odd filling factors at zero displacement field, respectively. That means there are some gapped states in these filling factors, indicating the possible formation of magneto-exciton condensates. The observation of the sequence of magneto-exciton condensates is quite novel, since the magneto-exciton condensates can only be found at total filling factor in GaAs system. Moreover, the author analyzes the gaps of the sequence quantitatively, which are well consistent with theory. Although this is certainly an interesting work, some issues are still needed to be addressed before publication in Nature Communication.

We thank the reviewer for appreciating the quality and merits of our manuscript. We also appreciate the detailed comments from this reviewer that have helped to improve our manuscript. We address the comments in detail below.

1. The authors mention that 10° twisted angle helps to reduce the tunneling strength because of momentum mismatch. Other large twisted angles can also make momentum mismatch in two layers. Are there some similar sequences of states observed in twisted double bilayer graphene with other large twisted angles?

We thank the reviewer for the important question. Although the twist angles of our three devices in this manuscript are all around 10° , we believe that the appearance of magnetically induced interlayer excitonic states should occur as long as the twist angle of the TDBG exceeds 3° . In addition, referring to theoretical and experimental works (Phys. Rev. Lett. 106, 126802 (2011), and ref. 21), when the twist angle is less than 3° , the moiré correlation effects and the increase in the effective mass of carriers will affect the decoupling effect of TDBG. At this point, under a magnetic field, the wave functions of the upper and lower graphene layers are more likely to hybridize, thus affecting the stability of the ECs.

In fact, besides the three large-angle samples mentioned in the main text, we also fabricated two samples with twist angles between 2° and 3° . In the sample with a twist angle of 2.3° , under the same experimental conditions, we did not observe any signs of ECs or their appearance (Fig. R2a). However, in the sample with a twist angle of 2.8° , although we did not observe distinct ECs, we could already see signs of ECs appearing (at $\nu_{\text{tot}}=-1$ marked by the red arrow in Fig. R2b).

Fig. R2. a: TDBG~ 2.3°

Fig. R2. b: TDBG~ 2.8°

2. What's the reason that magneto-exciton condensates are not observed in some odd filling factors? For example, there are no magneto-exciton condensates at and in the hole regime. Especially, it is contradictory to the theory that the system should have a large gap at since the orbital occupancy is .

We thank the reviewer for the important question. We do observe magneto-exciton condensates in the hole regime. We have observed gapped ECs at $\nu=-7, -1, 1, 3$ and 5 (red dotted circle in Fig. 2a and Fig. 2c). In the hole regime, states at $\nu=-7, -1$ have both longitudinal conductance dips, bulk resistance peaks and well-developed gaps, corresponding to exciton condensates (Fig 3c).

At $\nu=-3$, we observed a longitudinal conductance dip and a bulk resistance peak (see Fig. 2a and Fig. 2b). Our theory predicts the presence of gapped EC here, yet experimentally, we did not observe a well-developed gap at 1.55K in device A. We attribute this to some possible local unevenness. The temperature dependence curves actually show an onset of a trend towards EC gap formation at a base temperature of 1.55K (Fig. R3a below). At lower temperatures (which can suppress disorder effects) or under stronger magnetic field (which can provide a larger interaction energy scale and enhances the EC gap that overcomes disorder effects), we may observe EC at $\nu=-3$.

Due to device A is destroyed during measurement, we made another device included in FIG. S3d, the hole side of this new device revealed well-developed ECs at $\nu=-3$ with a distinct gap. (also see Fig. R3b, c below).

Fig. R3. (a). Temperature dependence of $R_{xx}-Bulk$ as a function of ν_{tot} at $D = 0$ V/nm and $B = 14$ T in device A. At $\nu_{tot} = -3$, the temperature dependence curves actually show an onset of a trend towards EC gap formation at a base temperature of 1.55K . **(b).** Temperature dependence of longitudinal conductance σ_{xx} as a function of ν_{tot} at $D = 0$ V/nm and $B = 14$ T in device C. The state at $\nu_{tot} = -3$ has a well-developed EC gap shown in **(c)**.

As for $\nu=-5$, according to theoretical analysis, ECs at $\nu=-5$ ($N=0$) exhibit a much smaller gap compared to $\nu=-1$ ($N=1$). It is possible that we may require extremely low temperatures and higher magnetic fields to induce a gap in this state. A similar situation applies to the state at $\nu=7$.

3. In Figure 3e, the authors show $R_{xx}-Bulk$ as a function of D to illustrate the evolution from exciton condensates to integer quantum Hall states where the red line shows the phase boundary. However, the clarification of how to define the boundary should be added in the main text, which may help the readers.

We thank the reviewer for the suggestion. we have added a description about how to define the boundary of ECs region and IQH region in both main text and caption in the revised manuscript.

‘At $\nu_{\text{tot}}=1$ in $N=0$ LL, red curve corresponding to meron–antimeron-type ECs with larger slope than gray curve corresponding to IQH, and at $\nu_{\text{tot}}=1$ in $N=1$ LL, red curve corresponding to particle–hole-type ECs with smaller slope than gray curve corresponding to IQH. (The rate of change of the bulk resistance with the displacement field near $D=0$ (the magnitude of the slope) determines the phase boundaries of different phases.)’

Previous numerical study (Physical review letters 79, 1718 (1997)) found that the excitation energy of meron–antimeron pairs show a sharp increase in gap with layer imbalance, while particle–hole excitation energy is independent of the layer imbalance until the zeeman energy exceeds the EC gap. In our system, layer imbalance is regulated by D . In this scenario, we find a sharp R_{xx} –Bulk decrease with D in red curve region for $\nu_{\text{tot}}=1$ whereas at $\nu_{\text{tot}}=5$, R_{xx} –Bulk hold a mild response with D (Fig. 3e). This suggests that ECs on electron side in $N=0$ LL host the spin-texture charged excitation while ECs in $N=1$ LL have a particle–hole excitation. In $N=0$ LL, red curve corresponding to meron–antimeron-type ECs with larger slope than gray curve corresponding to IQH, and in $N=1$ LL, red curve corresponding to particle–hole-type ECs with smaller slope than gray curve corresponding to IQH. For clarity, we mention the slopes of different phase to show the boundaries (Fig. R4).

Fig. R4. R_{xx} –Bulk as a function of D for the ECs at $\nu_{\text{tot}}=1, 5$, which correspond to $N=0$ and 1 LL, respectively. Red curve regions mark ECs and grey curves correspond to IQH regime, The gray dashed line is an extension of the gray solid line.

4. The axis label of Figure 3e should be changed from R_{xx} -bulk.

We thank the reviewer for noticing this typo. We have corrected this error in the revision.

Reviewer #3 (Remarks to the Author):

This work reports the result of experimental transport studies and theoretical calculations of the twisted bilayer graphene heterostructure with the graphene layers aligned at a high angle. Similar heterostructures were explored before, however, they either have a spacer between the graphene

layers, which leads to large separation between the layers and lower interlayer interactions or are unipolar in nature and only report observations on either hole or electron side.

In my view, the topic manuscript is suitable for Nature Communications as it reports on the observation of an exciton condensate in a novel two-dimensional electronic system with ultrastrong interlayer interactions due to the absence of a spacer layer.

We thank the reviewer for the time to read our manuscript and for the positive comments.

While authors cite several other works that study the same type of heterostructure, the authors are the first to observe a complete sequence of a strong Bose–Einstein exciton condensate at both electron and hole fillings. A few remarks:

1. What allows the authors to study the complete sequence of EC on both electron and hole fillings as opposed to the research done in references [16] and [17]? Is it a better quality of devices or a difference in the measurement technique? A discussion explaining the cause of the differences in [16,17] and the authors work would be very instructive.

We thank the reviewer for raising the valid point and the helpful suggestion. We have supplemented the relevant emphasis about the cause of the differences in [16,17] and our work in the revision.

‘Although recent studies have shown some plausible traces of ECs in such twisted bilayers by observing quantum Hall states (QHSs) at some incomplete odd-integer total fillings limited on one carrier side [16,17], due to contact quality issues and disorders [16], the sequence of ECs is yet to be observed.’

‘Here, we reported the observation of a complete sequence of ECs emerging at both electron and hole fillings with extremely strong coupling strength in high quality large twisted angle twisted double bilayer graphene(TDBG) devices’

We believe that the more comprehensive sequence of magneto-exciton condensates that emerging on both electron and hole sides in our work, is due to the higher quality and cleanliness of our devices compared to references 16 and 17. Besides, for ref.17, the structural differences in the Landau level between bilayer and monolayer graphene, as well as the asymmetry in the electronic-hole symmetry structures, may also contribute to yielding different outcomes. In addition to the potentially more favourable electronic dispersion in comparison with monolayer, the zeroth Landau level of bilayer is eightfold degenerate, with the spin and valley isospin degeneracy supplemented by an accidental orbital degeneracy. This multitude of broken symmetry states further expands the phase diagram of possible superfluid states.

For the measurement, our transport measurements were carried out in cryogenic superconducting magnets with base temperature of 1.5K. The four-terminal resistance were measured using low-frequency lock-in techniques at 17.777Hz with a current excitation of 20nA.

For the Ref.17, they employed a standard lock-in technique with $I = 100$ nA and $f = 13.333$ Hz for the twisted bilayer graphene device. Magnetotransport measurements were taken in two different cryostats; a He4 dry cryostat (JANIS) and a Leiden dilution refrigerator with a base

temperature of 30 mK. We think that these measurement difference cannot cause the differences of our work between Ref. 16,17.

| 2. In the description of Figure 1f, the x axis is described as n while the figure itself shows v_{tot}

We thank the reviewer for noticing this mistake. We have corrected this error by changing the n in the caption to the nu_{tot} in the revision.

REVIEWERS' COMMENTS

Reviewer #1 (Remarks to the Author):

The authors have satisfactorily addressed my previous questions and comments. I recommend the publication of the manuscript in Nature Communications.

Reviewer #2 (Remarks to the Author):

I have read the responses of the authors and the revised manuscript. I am satisfied with the new manuscript and recommend it to be accepted for publication in Nature Communications.

Reviewer #3 (Remarks to the Author):

In the revised version of the manuscript, the authors expanded on the cause of differences between their work and refs [16, 17]. They also addressed minor mistakes in figure captions. Since the authors have satisfactorily responded to all the comments, I recommend publication of this article.